# Diffusion Models with Deterministic Normalizing Flow Priors

**Mohsen Zand[1,2]***, **Ali Etemad[2], Michael Greenspan[2]**

[1]*Research Computing Center, University of Chicago*
[2]*Department of Electrical and Computer Engineering, and Ingenuity Labs Research Institute, Queen's University*

*zand@uchicago.edu, ali.etemad@queensu.ca, michael.greenspan@queensu.ca*

**Reviewed on OpenReview:** *https://openreview.net/forum?id=ACMNVwcR6v*

## Abstract

For faster sampling and higher sample quality, we propose DiNof (**Di**ffusion with **No**rmalizing **f**low priors), a technique that makes use of normalizing flows and diffusion models. We use normalizing flows to parameterize the noisy data at any arbitrary step of the diffusion process and utilize it as the prior in the reverse diffusion process. More specifically, the forward noising process turns a data distribution into partially noisy data, which are subsequently transformed into a Gaussian distribution by a nonlinear process. The backward denoising procedure begins with a prior created by sampling from the Gaussian distribution and applying the invertible normalizing flow transformations deterministically. To generate the data distribution, the prior then undergoes the remaining diffusion stochastic denoising procedure. Through the reduction of the number of total diffusion steps, we are able to speed up both the forward and backward processes. More importantly, we improve the expressive power of diffusion models by employing both deterministic and stochastic mappings. Experiments on standard image generation datasets demonstrate the advantage of the proposed method over existing approaches. On the unconditional CIFAR10 dataset, for example, we achieve an FID of 2.01 and an Inception score of 9.96. Our method also demonstrates competitive performance on CelebA-HQ-256 dataset as it obtains an FID score of 7.11. Code is available at https://github.com/MohsenZand/DiNof.

## 1 Introduction

Diffusion models (Sohl-Dickstein et al., 2015; Dhariwal & Nichol, 2021; Chen et al., 2020; Ho et al., 2020; Kong et al., 2020; Song & Ermon, 2020) are a promising new family of deep generative model that have recently shown remarkable success on static visual data, even beating GANs (Generative Adversarial Networks) (Goodfellow et al., 2014) in synthesizing high-quality images and audio. In a diffusion model, noise is gradually added to the data samples using a predetermined stochastic forward process, converting them into simple random variables. This procedure is reversed using a separate backward process that progressively removes the noise from the data and restores the original data distributions. In particular, the deep neural network is trained to approximate the reverse diffusion process by predicting the gradient of the data density.

Existing diffusion models (Yang et al., 2022; Zhang & Chen, 2021; Wehenkel & Louppe, 2021) define a Gaussian distribution as the prior noise, and a non-parametric diffusion method is developed to procedurally convert the signal into the prior noise. The traditional Gaussian prior is simple to apply, but since the forward process is fixed, the data itself has no impact on the noise that is introduced. As a result, the learned network may not model certain intricate but significant characteristics within the data distribution. Particularly, employing purely stochastic prior noise in modeling complex data distributions may not fully leverage the information and completely encompass all data details in the diffusion models.

---

*This work was partially done when Mohsen Zand was a Postdoctoral Fellow at Queen's University.

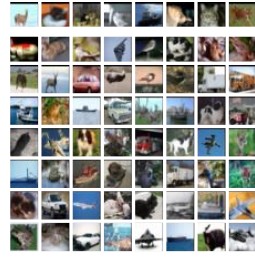
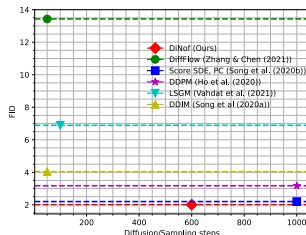

Figure 1: Uncurated samples generated by DiNof on CelebA-HQ-256 (left) and CIFAR-10 (middle) datasets. On the right figure, sample quality in terms of FID is shown versus diffusion/sampling steps for different diffusion-based generative models. As opposed to several other methods, DiNof can speed up the process while improving the sample quality.

Another issue is that the sampling procedure in most current methods involves hundreds or thousands of steps (time-discretization stages) (Salimans & Ho, 2022; Ramachandran et al., 2017; Zhang & Chen, 2021). This is due to the fact that the noise in the forward process must be added to the data at a slow enough rate to enable a successful reversal of the forward process for the reverse diffusion to produce high-quality samples. This, however, slows down training and sampling because it requires sufficiently long trajectories (the paths between the data space and the latent space), resulting in a substantially slower sampling rate than, for example, GANs or VAEs (Lu et al., 2022; Zhang & Chen, 2022; Salimans & Ho, 2022), which are single-step at inference.

In this work, we leverage the use of flow-based models, to learn noise priors deterministically from the data itself, which is then applied to improve the effectiveness of diffusion-based modelling. We propose the use of a deterministic prior as an alternative to the completely random noise of conventional diffusion models. We specifically develop a novel diffusion model where the data and latent spaces are connected by the nonlinear invertible maps from normalizing flows. Our model thus preserves all the benefits of the original diffusion model formulation, while employing both deterministic and stochastic trajectories in the mappings between the latent and data spaces. Our hypothesis is that data-informed latent feature representations and using both stochastic and deterministic processes can improve the representational quality and sample fidelity of generative models. We can generate samples using fewer sampling steps while attaining a sample quality that is superior to existing models (see Figure 1). Moreover, our model can be used in both conditional and unconditional settings, which can enable various image and video generation tasks.

Our contributions can be summarized as follows:

(**1**) We propose the use of a data-dependent, deterministic prior as an alternative to the random noise used in standard diffusion models, to enable modeling complex distributions more accurately, with a smaller number of sampling steps. Our new generative model leverages the strengths of both diffusion models and normalizing flows to improve both accuracy and efficiency in the mappings between the latent and data spaces.

(**2**) We evaluate our approach on CIFAR-10 and CelebA-HQ-256 datasets, which are the most commonly used datasets in this problem space. We achieve competitive results in the image generation task, yielding 2.01 and 7.11 FID scores on the CIFAR-10 and Celeb datasets, respectively.

(**3**) To allow reproducibility and contribute to the area, we release our code at:

https://github.com/MohsenZand/DiNof.

## 2 Related Work

There has been previous research looking towards creating a more informative prior distribution for deep generative models. For instance, hand-crafted priors (Nalisnick & Smyth, 2016; Tomczak & Welling, 2018), vector quantization (Razavi et al., 2019), and data-dependent priors (Li et al., 2020; Lee et al., 2021) have

been proposed in the literature. As priors for variational autoencoders (VAE), normalizing flows and hierarchical distributions (Maaløe et al., 2019; Child, 2020; Vahdat & Kautz, 2020; Rezende & Mohamed, 2015; Kingma et al., 2016) have been employed in particular. Prior distributions have also been defined implicitly (Bauer & Mnih, 2019; Aneja et al., 2020; Takahashi et al., 2019). Mittal et al. (2021) parameterized the discrete domain in the continuous latent space for training diffusion models on symbolic music data. They conducted two independent stages of training for a VAE and the denoising diffusion model. Wehenkel & Louppe (2021) proposed an end-to-end training method that modeled the prior distribution of the latent variables of VAEs using Denoisng Diffusion Probabilistic Models (DDPMs). They demonstrated that the diffusion prior model outperformed the Gaussian priors of traditional VAEs and was competitive with normalized flow-based priors. They also showed how hierarchical VAEs might profit from the improved capability of diffusion priors. Sinha et al. (2021) combined contrastive learning with diffusion models in the latent space of VAEs for controllable generation.

More recently, Lee et al. (2021) proposed PriorGrad to improve conditional denoising diffusion models with data-dependent adaptive priors for speech synthesis. They calculated the statistics from conditional data and utilized them as the Gaussian prior's mean and variance. Vahdat et al. (2021) proposed the Latent Score-based Generative Model (LSGM), a VAE with a score-based generative model (SGM) prior. They used the SGM after mapping the input data to latent space. The distribution over the data set embeddings was then modeled by the SGM. New data synthesis was accomplished by creating embeddings by drawing data from a base distribution, iteratively denoising the data, and then converting the embedding to data space via a decoder.

Denoising Diffusion Implicit Model (DDIM) proposed by Song et al. (2020a) was basically a fast sampling algorithm for DDPMs. It creates a new implicit model with the same marginal noise distributions as DDPMs while deterministically mapping noise to images. Specifically, an alternative non-Markovian noising process is developed that has the same forward marginals as the DDPM but enables the generation of various reverse samplers by modifying the variance of the reverse noise. An implicit latent space is therefore created from the deterministic sampling process. This is equivalent to integrating an ordinary differential equation (ODE) in the forward direction, followed by obtaining the latents in the backward process to generate images. Salimans & Ho (2022) proposed to reduce the number of sample steps in a progressive distillation model. The knowledge of a trained teacher model, represented by a deterministic DDIM, is distilled into a student model with the same architecture but with progressively halved sampling steps, thereby improving efficiency.

Slow generating speed continues to be a significant disadvantage of diffusion models, despite the outstanding performance and numerous variations. Different strategies have been investigated to overcome the efficiency issue. Rombach et al. (2022) used pre-trained autoencoders to train a diffusion model in a low-dimensional representational space. The latent space learned by an autoencoder was employed for both the forward and backward processes. Deterministic forward and reverse sampling strategies were suggested by DDIM (Song et al., 2020a) to increase generation speed.

Similar to our approach, Zhang & Chen (2021) connected normalizing flows and diffusion probabilistic models, and presented diffusion normalizing flow (DiffFlow) based on stochastic differential equations (SDEs). The reverse and forward processes were made trainable and stochastic by expanding both diffusion models and normalizing flows. They extended the normalizing flow approach by progressively introducing noise to the sampling trajectories to make them stochastic. The diffusion model was also expanded by making the forward process trainable. By minimizing the difference between the forward and the backward processes in terms of the Kullback-Leibler (KL) divergence of the induced probability measures, the forward and backward diffusion processes were trained simultaneously. Kim et al. (2022) also combined a normalizing flow and a diffusion process. They proposed an Implicit Nonlinear Diffusion Model (INDM), which implicitly constructed a nonlinear diffusion on the data space by leveraging a linear diffusion on the latent space through a flow network. The linear diffusion was expanded to trainable nonlinear diffusion by combining an invertible flow transformation and a diffusion model. Recently, Zhang et al. (2023) developed UnifiedDiffFlow, a unified theoretic framework for score-based diffusion models and GANs. It enabled a flexible trade-off between high sample quality and fast sampling speed by utilizing neural networks to approximate the SDE dynamics.

Truncated diffusion probabilistic modeling (TDPM) (Zheng et al., 2023), an adversarial auto-encoder empowered by both the diffusion process and a learnable implicit prior, is another method in this vein. Similar to LSGM, TDPM utilized variational autoencoder when transitioning from data to latent space. Specifically, TDPM is most closely related to an adversarial auto-encoder (AAE) with a fixed encoder and a learnable decoder, which uses a truncated diffusion and a learnable implicit prior. It is therefore a diffusion-based AAE that emphasizes shortening the diffusion trajectory through learning an implicit generative distribution.

Another related approach that addresses the issue of sampling from diffusion models to produce high-quality images is the diffusion probabilistic model based on optimal transport (DPM-OT) (Li et al., 2023). DPM-OT conceptualizes the process of inverse diffusion as an optimal transport (OT) problem, focusing on the transition between latent representations at different stages.

In contrast to existing techniques, our method uses SDEs and ODEs to map between data space and latent space utilizing both linear stochastic and nonlinear deterministic trajectories. We concentrate on *nonlinearizing* the diffusion process by using *nonlinear trainable deterministic* processes via *nonlinear invertible flow* mapping.

## 3  Method

Although diffusion models have unique advantages over other generative models, including stable and scalable training, insensitivity to hyperparameters, and mode-collapsing resilience, producing high-fidelity samples from diffusion models involves a fine discretization sampling process with sufficiently long stochastic trajectories (Song et al., 2020b; Vahdat et al., 2021; Zhang & Chen, 2021; Zhang et al., 2022). Several works investigated this and improved the efficiency. However, their improved runtime comes at the expense of poorer performance compared to the seminal work by Song et al. (2020b).

This motivates us to develop a new approach that simultaneously improves the sample quality and sampling time. We hence propose to integrate nonlinear deterministic trajectories in the mapping between the data and latent spaces. The deterministic trajectories are learned by using normalizing flows. In our diffusion/sampling steps, therefore, both stochastic and deterministic trajectories are employed. In the following subsections, we provide preliminary remarks on diffusion models and normalizing flows before introducing our approach.

### 3.1  Background

#### 3.1.1  Diffusion Models

Diffusion models are latent variable models that represent data $x(0)$ through an underlying series of latent variables $\{x(t)\}_{t=0}^{T}$. The key concept is to gradually destroy the structure of the data $x(0)$ by applying a diffusion process (*i.e.*, adding noise) to it over the course of $T$ time steps. The incremental posterior of the diffusion process generates $x(0)$ through a stochastic denoising procedure (Yang et al., 2022; Rasul et al., 2021; Voleti et al., 2022; Ho et al., 2022; Croitoru et al., 2022; Ho et al., 2020; Song et al., 2020a).

The diffusion process (also known as the forward process) is not trainable and is fixed to a Markov chain that gradually adds Gaussian noise to the signal. For a continuous time variable $t \in [0, T]$, the forward diffusion process $\{x(t)\}_{t=0}^{T}$ is defined by an *Itô* SDE as:

$$\mathrm{dx} = \mathbf{f}(\mathrm{x}, t)\mathrm{d}t + g(t)\mathrm{dw}, \tag{1}$$

where $\mathbf{f}(., t) : \mathbb{R}^d \to \mathbb{R}^d$ and $g(.) : \mathbb{R} \to \mathbb{R}$ denote a drift term and diffusion coefficient of $x(t)$, respectively. Also, w denotes the standard Wiener process (known as Brownian motion). To obtain a diffusion process as a solution for this SDE, the drift coefficient should be chosen so that it gradually diffuses the data $x(0)$, while the diffusion coefficient regulates the amount of added Gaussian noise. The forward process transforms $x(0) \sim p_0$ into simple Gaussian $x(T) \sim p_T$ so that at the end of the diffusion process, $p_T$ is an unstructured prior distribution that contains no information of $p_0$, where $p_t(x)$ denotes the probability density of $x(t)$.

It is shown by Song et al. (2020b) that the SDE in Eq. 1 can be converted to a generative model by starting from samples of $x(T) \sim p_T$ and reversing the process as a reverse-time SDE given by:

$$\mathrm{dx} = [\mathbf{f}(\mathrm{x}, t) - g^2(t)\nabla_\mathrm{x} \log p_t(\mathrm{x})]\mathrm{d}t + g(t)\mathrm{d}\bar{\mathrm{w}}, \tag{2}$$

where $\bar{\mathrm{w}}$ denotes a standard Wiener process when the time is reversed from $T$ to $0$ and $\mathrm{d}t$ denotes an infinitesimal negative time step. We can formulate the reverse diffusion process from Eq. 1 and simulate it to sample from $p_0$ after determining the $\nabla_\mathrm{x} \log p_t(\mathrm{x})$ score for each marginal distribution for all $t$. We can thereby restore the data by eliminating the drift that caused the data destruction.

A time-dependent score-based model $s_\theta(\mathrm{x}, t)$ can be trained to estimate $\nabla_\mathrm{x} \log p_t(\mathrm{x})$ at time $t \sim \mathcal{U}[0, T]$ by optimizing the following objective:

$$\theta^* = \arg\min_\theta \mathbb{E}_t\{\lambda(t)\mathbb{E}_{\mathrm{x}(0)}\mathbb{E}_{\mathrm{x}(t)|\mathrm{x}(0)}[\|s_\theta(\mathrm{x}(t), t) - \nabla_{\mathrm{x}(t)} \log p_{0t}(\mathrm{x}(t)|\mathrm{x}(0))\|_2^2]\}, \tag{3}$$

where $\lambda : [0, T] \to \mathbb{R}^+$ denotes a weighting function, $\mathrm{x}(0) \sim p_0(x)$, $\mathrm{x}(t) \sim p_{0t}(\mathrm{x}(t)|\mathrm{x}(0))$, and $p_{0t}$ denotes the transition from $\mathrm{x}(0)$ to $\mathrm{x}(t)$. By using score matching with enough data and model capabilities, the optimum solution $s_{\theta^*}(\mathrm{x}, t)$ can be achieved for nearly all x and $t$. It is hence equivalent to $\nabla_\mathrm{x} \log p_t(\mathrm{x})$.

To solve Eq. 3, the transition kernel $p_{0t}(\mathrm{x}(t)|\mathrm{x}(0))$ must be known. For an affine drift coefficient $\mathbf{f}$, it is usually a Gaussian distribution, whose mean and variance are known in closed-forms. Once a time-dependent score-based model $s_\theta$ has been trained, it can be utilized to create the reverse-time SDE. It can then be simulated using numerical methods to generate samples from $p_0$. Any numerical method applied to the SDE specified in Eq. 1 can be used to carry out the sampling. Song et al. (2020b) introduced some new sampling techniques, the Predictor-Corrector (PC) sampler being one of the best at generating high-quality samples. In PC, at each time step, the numerical SDE solver is used as a predictor to give an estimate of the sample at the next time step. Then, a score-based technique, such as the annealed Langevin dynamics (Song & Ermon, 2019), is used as a corrector to correct the marginal distribution of the estimated sample. The annealed Langevin dynamics algorithm (Song & Ermon, 2019) begins with white noise and runs $\mathrm{x}_i = \mathrm{x}_{i-1} + \frac{\gamma}{2}\nabla_\mathrm{x} \log p(\mathrm{x}) + \sqrt{\gamma}\mathrm{w}_i$, a certain number of iterations, where $\gamma$ controls the magnitude of the update in the direction of the score $\nabla_\mathrm{x} \log p(\mathrm{x})$.

The reverse-time SDE can also be solved using probability flow, a different numerical approach, thanks to score-based models (Song et al., 2020b). A corresponding deterministic process with trajectories that have the same marginal probability densities $\{p_t(\mathrm{x})\}_{t=0}^T$ as the SDEs exists for every diffusion process. This process is deterministic and fulfills an ordinary differential equation (ODE) as (Song et al., 2020b):

$$\mathrm{dx} = [\mathbf{f}(\mathrm{x}, t) - 1/2\mathbf{G}(t)\mathbf{G}(t)^\top \nabla_\mathrm{x} \log p_t(\mathrm{x})]\mathrm{d}t. \tag{4}$$

This process is called probability flow ODE. Once the scores are known, one can determine it from the SDE.

### 3.1.2 Normalizing Flows

In normalizing flows, a random variable with a known (usually Normal) distribution is transformed via a series of differentiable, invertible mappings (Abdelhamed et al., 2019; Kobyzev et al., 2020; Zhang & Chen, 2021; Zand et al., 2023). Let $\mathbf{Z} \in \mathbb{R}^D$ be a random variable with the probability density function $p_\mathbf{Z} : \mathbb{R}^D \to \mathbb{R}$ being a known and tractable function and $\mathbf{Y} = g(\mathbf{Z})$. The probability density function of the random variable $\mathbf{Y}$ can then be calculated using the change of variables formula as shown below:

$$p_\mathbf{Y}(y) = p_\mathbf{Z}(f(y))|\det \frac{\partial f}{\partial y}| = p_\mathbf{Z}(f(y))|\det \frac{\partial g}{\partial f(y)}|^{-1}, \tag{5}$$

where $\frac{\partial g}{\partial f(y)}$ denotes the Jacobian of $f$, and $f$ is the inverse of $g$.

In generative models, the aforementioned function $g$ (a generator) 'pushes forward' the initial base density $p_\mathbf{Z}$, often known as the 'noise', to a more complicated density. This is the generative direction, in which a data point $y$ is generated by sampling $z$ from the base distribution and applying the generator as $y = g(z)$. In order to normalize a complex data distribution, the inverse function $f$ moves (or 'flows') in the opposite

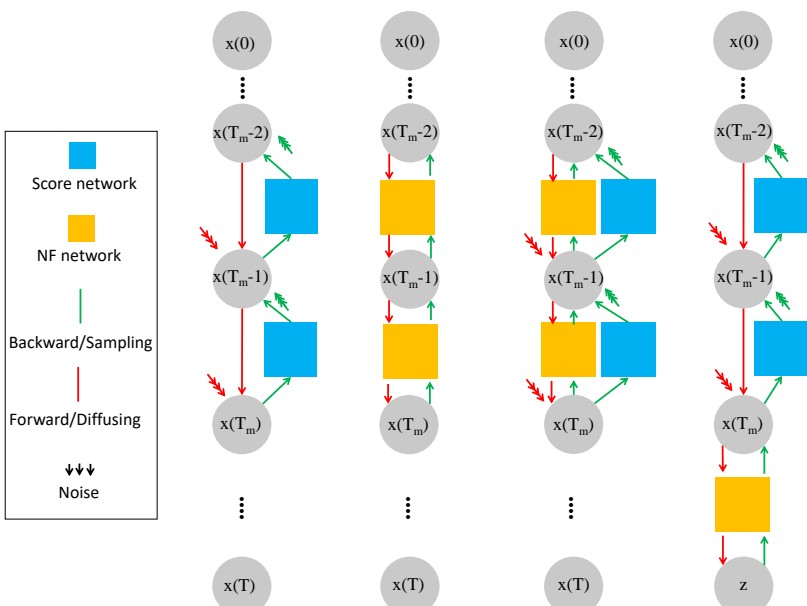

Figure 2: Architectural comparison of Diffusion, Normalizing Flows, DiffFlow (Zhang & Chen, 2021), and the proposed DiNof models. DiffFlow uses stochastic and trainable processes for both the forward and the backward processes, whereas DiNof utilizes a deterministic trainable process only at the final steps of the forward process. The backward process initiates with a deterministic process and turns to a stochastic process to generates images. We use $T_m$ to denote an arbitrary intermediate latent variable between data space and latent space.

way, from the complex distribution to the simpler, more regular or 'normal' form of $p_{\mathbf{Z}}$. Given that $f$ is 'normalizing the data distribution', this viewpoint is the source of the term 'normalizing flows'.

It can be challenging to build arbitrarily complex nonlinear invertible functions (bijections) such that the determinant of their Jacobian can be calculated. To address this, one strategy is to use their composition, since the composition of invertible functions is itself invertible. Let $g_1, \ldots, g_M$ be a collection of $M$ bijective functions, and let $g = g_M \circ g_{M-1} \circ \cdots \circ g_1$ represent the composition of the functions. The function $g$ can then be demonstrated to be bijective as well, which its inverse given as $f = f_1 \circ \cdots \circ f_{M-1} \circ f_M$. The latent variables are then given as:

$$
\begin{aligned}
\mathrm{x}_i &= f_i(\mathrm{x}_{i-1}, \theta) \\
\mathrm{x}_{i-1} &= f_i^{-1}(\mathrm{x}_i, \theta),
\end{aligned}
\tag{6}
$$

where $\{\mathrm{x}_i\}_{i=0}^M$ denote the trajectories between the data space and the latent space.

## 3.2 Proposed Method

A schematic illustration of the proposed method in comparison to other generative models is shown in Figure 2. In diffusion models, the forward process is fixed while the backward process is trainable. Yet, they are both stochastic. Both the forward and the backward processes of normalizing flow are deterministic. They combine into a single process since they are the inverse of one another. In DiffFlow (Zhang & Chen, 2021), both the forward and the backward processes are stochastic and trainable. Our method however employs both stochastic and deterministic trajectories that follow each other in both directions. This is more effective due to the possible reduction of the diffusion/sampling steps.

More specifically, we aim to improve the effectiveness of diffusion-based modelling by representing data $\mathrm{x}(0)$ via a set of latent variables $\mathrm{x}(t)$ between the data distribution and a data-dependent and deterministic prior

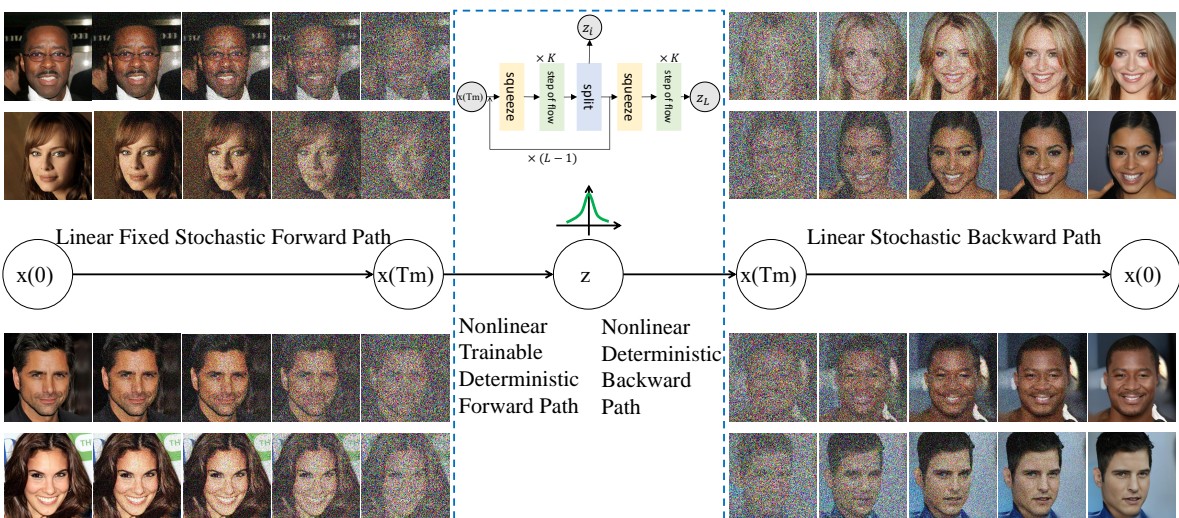

Figure 3: An overview of DiNof. It employs both linear stochastic and nonlinear deterministic trajectories in the mapping between data space and latent space using SDEs and ODEs. It hence utilizes normalizing flows to nonlinearize the diffusion models. Glow (Kingma & Dhariwal, 2018) architecture is used as the normalizing flow model.

distribution. We use a total number of $N$ noise scales and define $\mathrm{x}(\frac{i}{N}) = \mathrm{x}_i$, where $\{\mathrm{x}_i\}_{i=0}^{N}$ denotes the Markov chain. An SDE and an ODE both are used to model the forward diffusion process. We employ an *Itô* stochastic SDE as follows:

$$\mathrm{d}\mathrm{x} = f(t)\mathrm{x}\mathrm{d}t + g(t)\mathrm{d}\mathrm{w}, \tag{7}$$

where $t$ is a continuous time variable uniformly sampled over $[0, T_m)$. Theoretically $T_m$ can be any number between $1$ and $T$, implying a potential for reducing the diffusion steps. Furthermore, we model latent variables $\{\mathrm{x}(t)\}_{t < T_m}$ in the forward process using Eq. 7. Here, we can use one of the original diffusion models, such as VESDE (variance exploding SDE) or VPSDE (variance preserving SDE) (Song et al., 2020b). These models are linear, where $\mathbf{f}(\mathrm{x}, t) = f(t)\mathrm{x}$ is a function of $\mathrm{x}(t)$, and $g$ is a function of $t$.

In VESDE, $f(t) = 0$ and $g(t) = \sqrt{d\sigma^2/dt}$, where $\sigma^2(t)$ denotes the variance of latent variable $\mathrm{x}(t)$. Comparably, $f(t) = -1/2\beta(t)$, and $g(t) = \sqrt{\beta(t)}$ in VPSDE, where $\beta(\frac{i}{N}) = \bar{\beta}_i$, and $\{\bar{\beta}_i = N\beta_i\}_{i=0}^{N}$. We can also employ the discretized versions of the VESDE and VPSDE known as SMLD and DDPM noise perturbations (Song et al., 2020b). Using these conventional linear SDEs in the forward diffusion process, we transform the data $\mathrm{x}(0) \sim p_0$ to a diffused distribution $p_{T_m}$. Hence, we connect the data space and the latent variables $\{\mathrm{x}(t)\}_{t < T_m}$ through stochastic trajectories.

We propose exploiting the nonlinearity in the diffusion process by applying a nonlinear ODE on the rest of trajectories (*i.e.*, $\{\mathrm{x}(t)\}_{t \geq T_m}$). In contrast to a few prior works that use nonlinearity in the diffusion models (Vahdat et al., 2021; Zhang & Chen, 2021; Kim et al., 2022), we follow the existing linear process with a subsequent nonlinear process. This is shown in Figure 3, where the diffusion process is nonlinearized by employing nonlinear trainable deterministic processes. Intuitively and empirically, utilizing both linear and nonlinear processes can boost the sample quality. As demonstrated in our experiments, executing an efficient nonlinear process after an existing linear process can reduce the number of sampling steps and accelerate sampling. We choose normalizing flows as the means to nonlinearize the diffusion models, as they learn the nonlinearity by invertible flow mapping. Normalizing flows are in this way used to complete the remaining steps of the diffusion process in a single phase, which improves efficiency.

In our normalizing flow network, we consider a bijective map between $p_{T_m}$ and $z$, a latent variable with a simple tractable density such as a Gaussian distribution as $p_\theta(z) = \mathcal{N}(z; 0, \mathbf{I})$. The log-likelihood of

$x = \mathrm{x}(T_m)$ is then defined as:

$$\log p_\theta(x) = \log p_\theta(z) + \log|\det(dz/dx)| = \log p_\theta(z) + \sum_i^M \log|\det(d\mathbf{h}_i/d\mathbf{h}_{i-1})| \tag{8}$$

where $\{\mathbf{h}_i\}_{i=1}^M$ are intermediate representations generated by the layers of a neural network, $\mathbf{h}_0 = \mathrm{x}(T_m)$, and $\mathbf{h}_M = z$. We train this model by minimizing the negative log-likelihood. The overall objective which includes training our SDE and our ODE is a joint training objective that merges the ODE objective with the diffusion model's score matching objective ( *i.e.*, Eqs. 3 and 8).

As noted earlier, data and the latent variables are coupled through both stochastic and deterministic trajectories in an end-to-end network. We therefore employ two different forms of trajectories for the mapping between data and latent spaces using SDEs and ODEs. Stochastic trajectories are utilized between $T_m$ latent variables, while deterministic trajectories are employed for $M$ latent variables in the flow process. Although $T_m + M$ might be greater than the $N$ steps of a standard diffusion model, employing an efficient flow network which generates samples at a single step will nevertheless, speed up the process (Song et al., 2020b).

The typical strategy in current diffusion models is to restore the original distribution by learning to progressively reverse the diffusion process, step by step, from $T$ to 0. In our approach, however, we reconstruct $\mathrm{x}(T_m)$ using the backward process in the flow network via a single path. Furthermore, by reconstructing $\mathrm{x}(T_m)$, the Gaussian noise is deterministically mapped to a partially noisy sample $p_{T_m}$. New samples are generated in the backward process by simulating remaining stochastic trajectories from $t = T_m$ to $t = 0$ by the reverse-time SDE.

In contrast to generic SDEs, we have extra information close to the data distribution that can be leveraged to improve the sample quality. Specifically, $p_{T_m}$ which is sampled by the flow network is used as an informative prior for the reverse-time diffusion. To incorporate the reverse-time SDE for sampling, we can use any general SDE solver. In our experiments, we choose stochastic samplers such as Predictor-Corrector (PC) to incorporate stochasticity in the process, which has been shown to improve results (Song et al., 2020b). Another notable advantage of this approach is its ability to semantically modify images by changing the value of $T_m$. It can interpolate between deterministic and stochastic processes. In our experiments, we demonstrate the impact of the value of $T_m$ on sample quality.

## 4 Experiments

We conduct a systematic evaluation to compare the performance of our method with competing methods in terms of sample quality on image generation.

### 4.1 Protocols and Datasets

We show quantitative comparisons for unconditional image generation on CIFAR-10 (Krizhevsky et al., 2009) and CelebA-HQ-256 (Karras et al., 2017). We perform experiments on these two challenging datasets following the conventional experimental setup in the field (such as Kim et al. (2022); Salimans & Ho (2022); Song et al. (2020a)). We follow the experimental design of Ho et al. (2020); Song et al. (2020b), using the Inception Score (IS) (Salimans et al., 2016) and Frechet Inception Distance (FID) (Heusel et al., 2017) for comparison across models.

We use different SDEs such as VESDE, VPSDE, and sub-VPSDE to show our consistency with the existing approaches. The NCSN++ architecture is used as our VESDE model whereas DDPM++ architecture is utilized for VPSDE and sub-VPSDE models. PC samplers with one corrector step per noise scale are also used to generate the samples. As our normalizing flow model, we use the multiscale architecture Glow (Kingma & Dhariwal, 2018) with the number of levels $L = 3$ and the number of steps of each level $K = 16$. We also set the number of hidden channels to 256.

To ensure that the amount of neural network evaluations required during sampling is consistent with prior work (Ho et al., 2020; Jing et al., 2022; Vahdat et al., 2021; Song et al., 2020b), we set $T = 1000$ and $T = 1$

Table 1: Effect of varying $\{T_m\}_{T/10}^{T}$ on the CIFAR-10 dataset, where $T = 1$

| Model | $T_m$ | # samp. steps ↓ | IS ↑ | FID ↓ |
|---|---|---|---|---|
| NCSN++ cont. (Song et al., 2020b) | - | 1000 | 8.91 | 4.29 |
| DiNof | 0.1 | 100 | 5.29 | 46.60 |
| | 0.2 | 200 | 6.11 | 34.63 |
| | 0.3 | 300 | 7.50 | 21.32 |
| | 0.4 | 400 | 8.82 | 10.76 |
| | 0.5 | 500 | **9.41** | 3.94 |
| | 0.6 | 600 | 9.23 | **3.16** |
| | 0.7 | 700 | 9.13 | 3.29 |
| | 0.8 | 800 | 8.99 | 3.95 |
| | 0.9 | 900 | 9.21 | 3.76 |
| | 1.0 | 1000 | 8.87 | 4.18 |

Table 2: Generative performance on CIFAR-10 dataset

| Class | SDE | Type | Method | IS ↑ | FID ↓ |
|---|---|---|---|---|---|
| GAN | - | - | StyleGAN2-ADA (Karras et al., 2020) | 9.83 | 2.92 |
| | | | SNGAN + DGflow (Ansari et al., 2020) | - | 9.62 |
| | | | TransGAN (Jiang et al., 2021) | - | 9.26 |
| | | | StyleFormer (Park & Kim, 2022) | - | 2.82 |
| | | | StyleSAN-XL (Takida et al., 2023) | - | 1.36 |
| VAE | - | - | NVAE (Vahdat & Kautz, 2020) | - | 23.5 |
| | | | DCVAE (Parmar et al., 2021) | - | 17.9 |
| | | | CR-NVAE (Sinha & Dieng, 2021) | - | 2.51 |
| Flow | - | - | Glow (Kingma & Dhariwal, 2018) | - | 46.90 |
| | | | ResFlow (Chen et al., 2019) | - | 46.37 |
| | | | Flow++ (Ho et al., 2019) | - | 46.4 |
| | | | DenseFlow-74-10 (Grčić et al., 2021) | - | 34.9 |
| Diffusion | Linear | - | NCSN (Song & Ermon, 2019) | 8.87 | 25.32 |
| | | | NCSN v2 (Song & Ermon, 2020) | 8.40 | 10.87 |
| | | | NCSN++ cont. (deep, VE) (Song et al., 2020b) | 9.89 | 2.20 |
| | | | DDPM (Ho et al., 2020) | 9.46 | 3.17 |
| | | | DDIM (Song et al., 2020a) | - | 4.04 |
| | | | Distillation (Salimans & Ho, 2022) | - | 2.57 |
| | | | Subspace Diff. (NSCN++, deep) (Jing et al., 2022) | 9.94 | 2.17 |
| | Nonlinear | SBP | SB-FBSDE (Chen et al., 2021) | - | 3.01 |
| | | OT | DPM-OT (Li et al., 2023) | - | 2.92 |
| | | VAE-based | LSGM (FID) (Vahdat et al., 2021) | - | 2.10 |
| | | | LSGM (NLL) (Vahdat et al., 2021) | - | 6.89 |
| | | | TDPM ($T_{Trunc=99}$) (Zheng et al., 2023) | - | 2.83 |
| | | Flow-based | DiffFlow (Zhang & Chen, 2021) | - | 13.43 |
| | | | INDM (FID) (Kim et al., 2022) | - | 2.28 |
| | | | INDM (NLL) (Kim et al., 2022) | - | 4.79 |
| | | | DiNof (Ours) | 9.96 | 2.01 |

for discrete and continuous diffusion processes, respectively. The number of noise scales $N$ is however set to 1000 for both cases. Additionally, the number of conditional Langevin steps is set to 1. The Langevin signal-to-noise ratio for CIFAR-10 and CelebA-HQ-256 are fixed at 0.16 and 0.17, respectively. The default settings are fixed for all other hyperparameters based on the optimal parameters determined in (Song et al., 2020b; Jing et al., 2022). All details are available in the source code release.

Table 3: Generative performance on CelebA-HQ-256 dataset

| Method | FID ↓ |
|---|---|
| Glow (Kingma & Dhariwal, 2018) | 68.93 |
| NVAE (Vahdat & Kautz, 2020) | 29.76 |
| DDIM (Song et al., 2020a) | 25.60 |
| SDE (Song et al., 2020b) | 7.23 |
| D2C (Sinha et al., 2021) | 18.74 |
| LSGM (Vahdat et al., 2021) | 7.22 |
| TDPM Zheng et al. (2023) | 8.38 |
| RDM Teng et al. (2023) | 3.15 |
| DiNof (Ours) | 7.11 |

Table 4: Generative performance and sampling time on CIFAR-10

| Model | IS ↑ | FID ↓ | Time ↓ |
|---|---|---|---|
| DDPM++ | 9.64 | 2.78 | 43s |
| TDPM ($T_{Trunc=99}$, DDPM++) | 9.62 | 2.83 | 37s |
| DiNof (Glow, DDPM++) | 9.65 | 2.51 | 24s |
| DDPM++ cont. (VP) | 9.58 | 2.55 | 45s |
| DiNof (Glow, DDPM++ cont. (VP)) | 9.75 | 2.40 | 25s |
| DDPM++ cont. (sub-VP) | 9.56 | 2.61 | 44s |
| DiNof (Glow, DDPM++ cont. (sub-VP)) | 9.73 | 2.43 | 24s |
| NCSN++ | 9.73 | 2.45 | 97s |
| DiNof (Glow, NCSN++) | 9.85 | 2.25 | 56s |
| NCSN++ cont. (VE) | 9.83 | 2.38 | 97s |
| DiNof (Glow, NCSN++ cont. (VE)) | 9.87 | 2.12 | 56s |
| NCSN++ cont. (deep, VE) | 9.89 | 2.20 | 150s |
| DiNof (Glow, NCSN++ cont. (deep, VE)) | 9.96 | 2.01 | 90s |

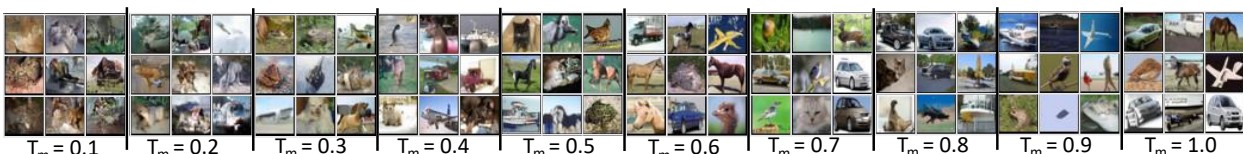

$T_m = 0.1$   $T_m = 0.2$   $T_m = 0.3$   $T_m = 0.4$   $T_m = 0.5$   $T_m = 0.6$   $T_m = 0.7$   $T_m = 0.8$   $T_m = 0.9$   $T_m = 1.0$

Figure 4: CIFAR-10 samples with different $T_m$ thresholds.

## 4.2 Results

### 4.2.1 Model Parameters

We first optimize for the CIFAR10 sample quality, and then we apply the resulting parameters to the other dataset. To find the optimal value for $T_m$, we investigate the results for a variety of thresholds. We select NCSN++ cont. (Song et al., 2020b), which is an NCSN++ model conditioned on continuous time variables as the baseline model. We calculate FIDs for various $T_m$ values with $T/10$ increments on the models trained for 500K training iterations with a batch size of 32, where $T = 1$. Depending on the $T_m$ value, a different number of sampling steps is used in our model. Note that the number of sampling steps are reduced except when $T_m = T$. In this case, normalizing flows provide an initial prior as an alternative to the standard Gaussian prior.

As reported in Table 1, the best IS and FID are obtained when $T_m = 0.5$ and $T_m = 0.6$, respectively. Also, our method with $T_m = 0.5$ and 500 fewer sampling steps, outperforms the baseline model. It achieves 0.35 improvement in terms of FID over the baseline method by obtaining an FID = 3.94. Improvements can be seen for all $T_m \geq 0.5$. For smaller $T_m$ values, however, our method suffers a significant degradation. This is due to the high and imbalanced stochasticity at smaller $T_m$ values. This is shown in Figure 4, where unrecognizable images are generated with a high stochasticity for $T_m < 0.5$. Smaller $T_m$ makes the backward process simple and fast but challenging to reconstruct the data. By additional noise, however, the high-quality images are successfully reconstructed, although with more sampling steps. Nonetheless, the capacity to explicitly trade-off between accuracy and efficiency is still a crucial feature. For instance, the number of function evaluations is decreased by nearly 50% while maintaining the visual quality of samples by using a smaller threshold, such as $T_m = 0.5$ that balances sample quality and efficiency (*i.e.*, the number of sampling steps). We fix $T_m = 0.6$ for the rest of the experiments as it results in the best FID score of 3.16.

### 4.2.2 Unconditional Color Image Generation

We compare our method with prominent diffusion-based models and nonlinear diffusions, providing a representative evaluation within the current state-of-the-art landscape. There have been a few prior works that

have leveraged diffusion models with nonlinear diffusions. Specifically, the examples found in the literature are: LSGM (Vahdat et al., 2021) which implements a latent diffusion using VAE; DiffFlow (Zhang & Chen, 2021) which uses a flow network to nonlinearize the drift term; SB-FBSDE (Chen et al., 2021; De Bortoli et al., 2021) which reformulates the diffusion model into a Schrodinger Bridge Problem (SBP); DPM-OT (Li et al., 2023) which treats inverse diffusion as an optimal transport (OT) problem between latents at various stages; TDPM (Zheng et al., 2023) that focuses on reducing the diffusion trajectory by learning an implicit generative distribution using AAE, and; INDM (Kim et al., 2022) which uses a flow network to implicitly construct a nonlinear diffusion on the data space by leveraging a linear diffusion on the latent space. We show that, compared to these methods, our flexible approach provides better generative modeling performance. We demonstrate this on the CIFAR-10 image dataset, which compared to CelebA-HQ-256 is far more diversified and multimodal.

As is done in previous research (Song et al., 2020b; Jing et al., 2022), the best training checkpoint with the smallest FID is utilized to report the results on CIFAR-10 dataset. One checkpoint is saved every 50k iterations for our models, which have been trained for 1M iterations. The batch size is also fixed to 128.

Table 2 summarizes the sample quality results on the CIFAR-10 dataset for 50K images. Our method with continuous NCSN++ (deep) obtains the best FID and IS of 2.01 and 9.96, respectively. The considerable improvement over other nonlinear models such as DiffFlow, SB-FBSDE, INDM, LSGM, and TDPM shows the potential of using both SDEs and ODEs. It is worth noting that although certain methods like DPM-OT Li et al. (2023) might require fewer steps in the reverse process, our method still maintains a competitive edge in terms of sample fidelity. More importantly, our method maintains full compatibility with the underlying diffusion models, and hence, retaining all their capabilities. Another key point is that our approach, in comparison to similar methods that integrate normalizing flows with diffusion models, like DiffFlow (Zhang & Chen, 2021), delivers notably better performance. The strength of our method lies in optimizing computational efficiency without compromising sample quality. With an FID score of 2.01, as opposed to DiffFlow's 13.43, our approach demonstrates that it can both accelerate the process and improve sample quality. Our method also consistently improves over the baseline methods. For example, it achieves a 0.19 improvement in the FID score compared to NCSN++ cont. (deep, VE) with the same diffusion architecture. Furthermore, our method not only preserves the flexibility of existing SDEs but also boosts their effectiveness.

We evaluate the applicability of our method to the high-resolution dataset of CelebA-HQ-256. We trained on this dataset for 0.5M iterations, and the most recent training checkpoint is used to derive the results. We use a batch size of 8 for training and a batch size of 64 for sampling. To save on computation, we use the optimal $T_m$ value of 0.6, which is obtained on the CIFAR-10 dataset. We also utilize the continuous NCSN++ with PC samplers. As shown in Table 3, DiNof achieves competitive performance in terms of FID on the CelebA-HQ-256 dataset. It specifically obtains a competitive FID of 7.11 which outperforms LSGM, another nonlinear model by a considerable margin of 0.11 FID. The performance gain is mainly contributed to the integrated deterministic nonlinear priors as our diffusion architecture is similar to the one used in SDE and LSGM.

### 4.2.3 Sampling Time

We evaluate DiNof in terms of sampling time on the CIFAR-10 dataset. We specifically measure the improved runtime in comparison to the original SDEs (VESDE, VPSDE, and sub-VPSDE) with PC samplers (Song et al., 2020b) on an NVIDIA A100 GPU. The sampler is discretized at 1000 time steps for all SDEs. We however discretize it at 600 steps in our models. Thus, we employ a considerably smaller number

Table 5: Intermediate results for different iteration numbers on the CIFAR-10 dataset

| Model | Iteration | IS ↑ | FID ↓ |
|---|---|---|---|
| DDPM++ | 50k | 8.62 | 6.05 |
| | 100k | 8.97 | 3.71 |
| | 150k | 9.34 | 2.99 |
| | 200k | 9.50 | 2.72 |
| | 250k | 9.57 | 2.70 |
| | 300k | 9.60 | 2.69 |
| DDPM++ cont. (VP) | 50k | 8.48 | 6.13 |
| | 100k | 8.96 | 3.83 |
| | 150k | 9.22 | 3.05 |
| | 200k | 9.43 | 2.79 |
| | 250k | 9.58 | 2.66 |
| | 300k | 9.71 | 2.69 |
| DDPM++ cont. (sub-VP) | 50k | 8.32 | 7.55 |
| | 100k | 8.75 | 4.91 |
| | 150k | 9.00 | 3.85 |
| | 200k | 9.15 | 3.35 |
| | 250k | 9.31 | 3.11 |
| | 300k | 9.37 | 2.94 |

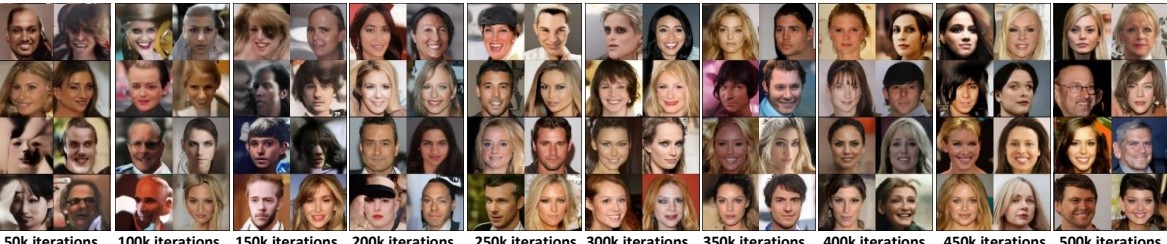

Figure 5: Visual samples for various iteration numbers during training on the CelebA-HQ-256 dataset.

of diffusion/sampling steps. As we employ a Glow architecture, our models include $\sim 7M$ more parameters than the standard SDE models. In contrast to the SDE models which are trained for 1.3M iterations, we train our models for 1M iterations to suppress overfitting. We follow the same strategy as (Song et al., 2020b), and report the results on the best training checkpoint with the smallest FID.

As shown in Table 4, our method consistently reduces sampling time and improves sample quality. For instance, it takes 24s to generate an image sample, while yielding an FID score of 2.51 on the DDPM++ model. It however takes 43s using the original DDPM++ architecture which achieves FID = 2.78. Our method is hence $\sim 1.7\times$ faster than original SDEs. It also performs better in terms of FID and IS. For instance, it improves over SDEs by $\sim 1.1\times$ FID on average.

### 4.2.4 Intermediate Results

We save one checkpoint every 50k iterations for our models and report the results on the best training checkpoint with the smallest FID. It is however worthwhile considering the intermediate results to better understand the entire training and inference procedures. In Table 5, we show IS and FID for DDPM++, DDPM++ cont. (VP), and DDPM++ cont. (sub-VP) models trained for various iteration numbers on the CIFAR-10 dataset. As can be observed, DDPM++ and DDPM++ cont. (VP) models improve more quickly than DDPM++ cont. (sub-VP).

Visual samples during training are also illustrated in Figure 5 for the CelebA-HQ-256 dataset, where improvements over the course of training are obvious. Further details are updated and modified as training progresses.

### 4.2.5 Qualitative Results

We visualize qualitative results for CelebA-HQ-256 and CIFAR-10 in Figure 1. We show that as opposed to several other methods, DiNof can speed up the process while improving the sample quality. Other methods that shorten the sampling process like (Song et al., 2020a; Zhang & Chen, 2021) frequently compromise the sample quality.

Random samples from our best models on the CIFAR-10 and CelebA-HQ-256 datasets are further depicted in Figure 6 and Figure 7, respectively. They demonstrate the robustness and reliability of our approach for generating realistic images. Our method creates diverse samples from various age and ethnicity groups on CelebA-HQ-256, together with a range of head postures and face expressions. DiNof also produces sharp and high-quality images with density details on the challenging multimodal CIFAR-10 dataset.

## 5 Conclusion

We propose DiNof, which improves sample quality while also reducing runtime. Our model breaks previous records for the inception score and FID for unconditional generation on both CIFAR-10 and CelebA-HQ-256 datasets. As compared to the previous best diffusion-based generative models, it is surprising that we are able to reduce the sampling time while improving the sample quality. Unlike many other methods, our approach also maintains full compatibility with the underlying diffusion models and so retains all their

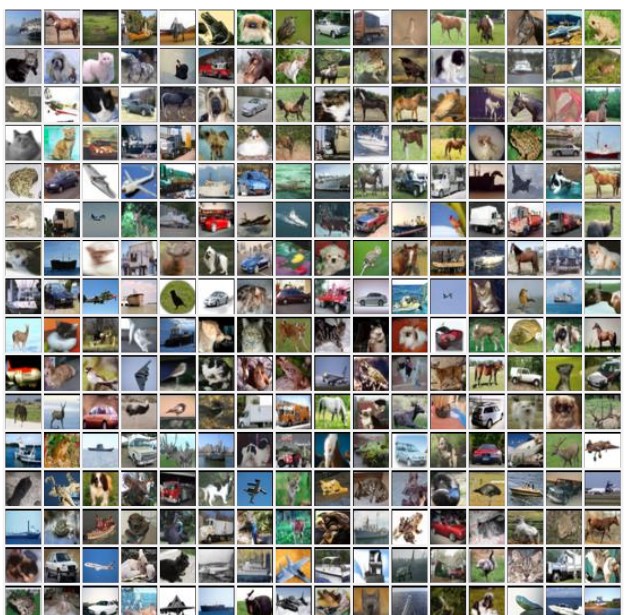 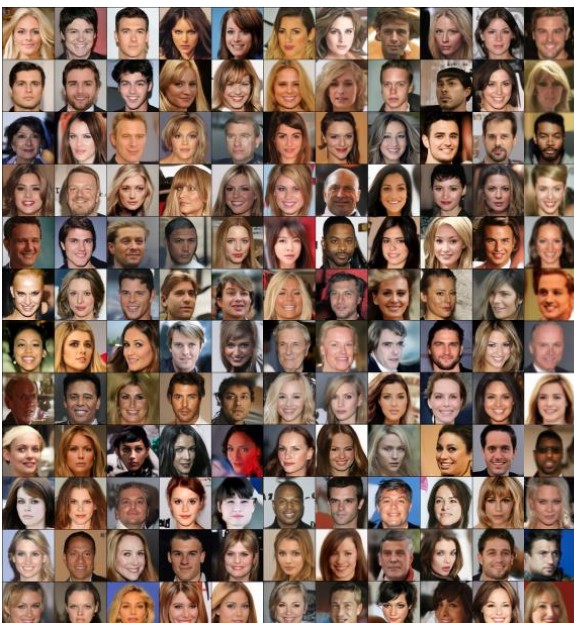

Figure 6: Uncurated CIFAR-10 generated samples. Figure 7: Uncurated CelebA-HQ-256 generated samples.

features. It should also be noted that even though incremental experiments are used to determine the optimal value of the single model hyperparameter $(T_m)$, we only needed to run 10 increments (on CIFAR-10) to beat the other SOTA methods and show how useful DiNof is in real-world situations. We did not conduct this experiment again to acquire our results on the CelebA-HQ-256 dataset, further demonstrating the robustness and applicability of our technique.

A variety of image and video generation tasks may be made possible by our model's ability to be applied in both conditional and unconditional scenarios. It is beneficial to investigate its effectiveness for several other tasks including conditional image generation, interpolations, colorization, and inpainting.

## Statement of Broader Impact

Our work enhances the efficacy of generative models in image generation, offering potential applications in creative tools, media accessibility, and computer vision research. However, these advancements carry risks, including the misuse of AI for creating misleading or harmful content, such as deepfakes, and amplifying biases in training data. We advocate for responsible AI use, adhering to ethical guidelines and transparency in deployment to mitigate these concerns.

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
