# OpenReview forum: "Diffusion Models with Deterministic Normalizing Flow Priors"
_TMLR — Accepted by TMLR_

### Review · Reviewer_GwNV · 2024-05-10

**Summary Of Contributions:**

This paper proposes to use a normalizing flow to speed up the inference process of a diffusion model. Specifically, it learns a normalizing flow to map from the distribution of $x(T)$ to that of $x(T_m)$ where $T_m <T$. In inference, it uses the normalizing flow to infer from $x(T)$ (i.e., a Gaussian noise) to $x(T_m)$ and then uses the diffusion model to further infer to an image.

**Audience:**

Yes

**Broader Impact Concerns:**

There are no concerns on the ethical implications of the work.

**Claims And Evidence:**

Yes

**Requested Changes:**

- More discussions to distinguish your work and DiffFlow (Zhang & Chen, 2021).
- Please discuss and compare with [1] in experiments.

**Strengths And Weaknesses:**

## Strengths
- The idea is simple and intuitive.
- The experimental results are good, indicating good FID and IS scores, while being faster than DDIM and DDPM.

## Weaknesses
- Actually, the idea is not novel, because there are some similar works [1] proposed to learn generators to map from $x(T)$ to $x(T_m)$. However, I do not make a decision mainly based on the novelty because novelty is not a main criterion to justify a TMLR paper.
- It would be great if the authors can discuss what the benefit of learning a normalizing flow to map from $x(T)$ to $x(T_m)$ rather a generator.

[1] Li Z, Li S, Wang Z, Lei N, Luo Z, Gu DX. DPM-OT: A New Diffusion Probabilistic Model Based on Optimal Transport. InProceedings of the IEEE/CVF International Conference on Computer Vision 2023 (pp. 22624-22633).

---

### Review · Reviewer_jU2g · 2024-07-15

**Summary Of Contributions:**

This paper proposes a new generative model, based on the joint fusion of a diffusion model and normalizing flow. The authors refer to their method as DiNof (which stands for Diffusion with Normalizing Flow Priors). In essence, the core proposition is the use of a data-dependent, deterministic normalizing flow prior as an alternative to the random noise currently used in the context of standard diffusion models. They claim this enables more accurate modelling of complex distributions and with a smaller number of sampling steps. These claims are validated on two datasets: CIFAR-10 and CelebA-HQ-256. DiNof attains FID scores of 2.01 and 7.11 on the CIFAR-10 and CelebA datasets, respectively.

**Audience:**

Yes

**Broader Impact Concerns:**

This work deals with increasing the efficacy of generative models. The potential ethical ramifications are similar to those of any modern generative AI method. A brief impact statement should be added to clarify this and highlight the potential adverse effects of generative AI.

**Claims And Evidence:**

No

**Requested Changes:**

As mentioned above in the weaknesses section, I believe the baselines in this paper are incomplete. In particular, the strongest baselines for CIFAR-10 and CelebA-HQ-256 from references [a] and [b] should be added to the paper, and all claims made regarding the state-of-the-art FID results should be rewritten/weakened.

In addition, a non-exhaustive list of minor corrections is given below:

- Section 1, third paragraph: "single-step at inferences" -> "single-step at inference"
- Section 2, first paragraph: "an VAE" -> "a VAE"
- Section 2, fifth paragraph: "a significant disadvantage of diffusion modeles" -> "a significant disadvantage of diffusion models"
- Section 3, second paragraph: "remarks on the diffusion models and normalizing flows" -> "remarks on diffusion models and normalizing flows"
- Section 3, final paragraph: "It can interpolate between deterministic and the stochastic processes." -> "It can interpolate between deterministic and stochastic processes."
- Section 4.2.2, first paragraph: Please use \citet for in-text citations (e.g. for De Bortoli et al., 2021).
- Section 4.2.2, third paragraph: "it achieves 0.19 increase in the FID" -> "it achieves a 0.19 decrease in the FID" or "it achieves a 0.19 improvement in the FID"
- Section 4.2.2, fourth paragraph: "To save the computations," -> "To save on computation,"
- Section 4.2.3, first paragraph: "Thus, we employ a considerably less number of diffusion/sampling steps" -> "Thus, we employ a considerably smaller number of diffusion/sampling steps"

**Strengths And Weaknesses:**

## Strengths

1. Using normalizing flow priors for diffusion models is something that has not been thoroughly investigated prior to this paper, and as such, some novelty is granted to this work. This augmentation helps reduce compute time considerably (as shown by Table 4) while attaining good performance on presented benchmarks.

## Weaknesses

1. Although this particular fusion of normalizing flows and diffusion models has not been investigated, the idea of using normalizing flows together with diffusion models has existed for some time; specifically since DiffFlow was presented by Zhang & Chen in 2021. To the paper's credit, they are upfront about this and introduce Figure 2 explicitly to distinguish their fusion from previously existing work. That being said, this does take away from some of the novelty, especially given that the fusion is not of a very complex nature (i.e. the paper is focused on presenting a diffusion model with a less trivial prior to reduce computational load).

2. The results for this paper are good, but the paper makes the misleading claim that several of the results are state-of-the-art. In particular, they claim that their approach "obtains a state-of-the-art FID of 7.11" on the CelebA-HQ-256 dataset. Although this is a solid result, there are at least 6 baselines with a better FID score than this (please see reference [a]). Some of these results date back to 2021, i.e., they are not that recent. Additionally, there are at least 20 baselines with a better FID than 2.01 on CIFAR-10 (please see reference [b]). The best of these baselines should be included in this paper for both results to avoid giving the reader a misleading representation about the strength of the method. All claims made about the state-of-the-art nature of the presented results should be weakened considerably.

## Verdict

This paper presents and investigates the fusion of using diffusion models with normalizing flow priors. This particular fusion of diffusion models and normalizing flows is novel, although the idea of combining diffusion models and normalizing flows has been studied before (Zhang & Chen, 2021). The paper's presented method performs well on the CIFAR-10 and CelebA-HQ-256 benchmarks, although the paper makes misleading claims about the state-of-the-art nature of their results. In particular, there are at least 6 papers with a better CelebA-HQ-256 result (see reference [a] below). That being said, the authors' approach does considerably reduce computational load while obtaining solid results. As such, I think this work would be interesting to some portion of TMLR's audience, however the claims made in the submission about the state-of-the-art nature of the results are not well supported and this language needs to be changed multiple times throughout the paper. Given that both criteria for acceptance to TMLR are not satisfied (in particular the papers' claims about their state-of-the-art results), I must recommend a reject rating for the paper in its current state.

### References

[a] Papers with Code: Image Generation on CelebA-HQ 256x256. https://paperswithcode.com/sota/image-generation-on-celeba-hq-256x256

[b] Papers with Code: Image Generation on CIFAR-10. https://paperswithcode.com/sota/image-generation-on-cifar-10

---

> ### Author Response · Authors · 2024-08-09
> **Response to reviewer jU2g**
>
> Thank you for your detailed review and valuable suggestions! We appreciate the opportunity to address your comments.
>
> > Although this particular fusion of normalizing flows and diffusion models has not been investigated, the idea of using normalizing flows together with diffusion models has existed for some time.
>
> While we acknowledge that the combination of normalizing flows and diffusion models has been previously explored, our method's strength lies in optimizing computational efficiency without compromising sample quality. Achieving a FID score of 2.01, compared to DiffFlow’s 13.43, our approach demonstrates that it can both accelerate the process and improve sample quality. We have clarified these points in the revised manuscript (Section 4.2.2, third paragraph).
>
> > The results for this paper are good, but the paper makes the misleading claim that several of the results are state-of-the-art. All claims made about the state-of-the-art nature of the presented results should be weakened considerably.
>
> We acknowledge the reviewer’s feedback and have revised the manuscript to accurately reflect the standing of our results relative to state-of-the-art methods.
> As suggested, we have included the strongest baselines for CIFAR-10 and CelebA-HQ-256 from the provided references in Table 2 and Table 3, respectively.
> Our method is designed to balance computational speed and sample quality, and we have revised the text to present our findings more precisely without overstating our claims.
>
> > Minor corrections
>
> As kindly suggested, we have corrected the typos and spelling in the revised manuscript.
>
> > A brief impact statement should be added to clarify this and highlight the potential adverse effects of generative AI.
>
> We have included the following impact statement in the revised manuscript (in a separate section after the conclusion):
>
> ``Our work enhances the efficacy of generative models in image generation, offering potential applications in creative tools, media accessibility, and computer vision research. However, these advancements carry risks, including the misuse of AI for creating misleading or harmful content, such as deepfakes, and amplifying biases in training data. We advocate for responsible AI use, adhering to ethical guidelines and transparency in deployment to mitigate these concerns.''

---

### Review · Reviewer_gvFE · 2024-07-16

**Summary Of Contributions:**

This paper proposed a new generative method named DiNof, which can be seen as a hybrid model between normalizing flows and diffusion models. The motivation is that the generative process of diffusion models can be speeded up by truncating the diffusion process into two stages: a linear diffusion to a noisy version of data distribution; then transforming the noisy data distribution into a prior distribution by normalizing flows. The experiments indicate the current approach can achieve a good trade-off between sample quality and sampling speed.

**Audience:**

Yes

**Broader Impact Concerns:**

N.A.

**Claims And Evidence:**

No

**Requested Changes:**

See Section: Strengths And Weaknesses.

**Strengths And Weaknesses:**

Pros:

- The idea of combining diffusion models with normalizing flows is new and the motivation in theory sounds plausible.

- There are broad evaluations on experiments to justify the effectiveness of the idea.

However, there are also evident cons that may be resolved to strengthen the claims and ideas of current paper.

Cons:

- The idea of combining diffusion models with other existing methods such as GANs, VAEs, or Normalizing flows to speed up training and sampling is not new. There are many existing works in both theoretical analysis and empirical validations to demonstrates this idea. For example, in [1], the authors show empirically [1] and theoretically [2] that the sampling process could be speeded up by truncating the diffusion process and learn the truncated prior by another generative model such as GANs or VAEs.
Also In [2], the authors demostrates that diffusion models can be unified into other existing generative models by adjusting the SDE/ODE, which is also relevent to the current theoretic motivations of DiNof.  The authors should make comprehensive discussions on the relevance and difference to existing works.

- The experiments are not persuasive enough to readers to demonstrate the superiority of current DiNof approach. In particular, the current approach shares plenty of motivations to [1], but the experimental results in table 3 Generative performance on CelebA-HQ-256 dataset and Table 4: Generative performance and sampling time on CIFAR-10, does not compare with [1]. I strongly suggest authors to add comparisons to [1] in every experiments.




References

[1]. Truncated Diffusion Probabilistic Models and Diffusion-based Adversarial Auto-Encoders.  https://openreview.net/forum?id=HDxgaKk956l

[2]. DiffFlow: A Unified SDE for Score-Based Diffusion Models and Generative Adversarial Network. https://openreview.net/forum?id=x17qiTPDy5

---

> ### Author Response · Authors · 2024-08-09
> **Response to reviewer gvFE**
>
> Thank you for your constructive review and thoughtful insights! We have made revisions to address the points you raised.
>
> > There are many existing works in both theoretical analysis and empirical validations to demonstrates this idea.
>
> Our work acknowledges the existing literature on combining diffusion models with other generative models to enhance efficiency.
> However, our approach offers a unique contribution:
>
> * Truncated diffusion probabilistic modeling (TDPM) utilizes variational autoencoders in transitioning from data to latent space.
>     It is most closely related to an adversarial auto-encoder (AAE) with a fixed encoder and a learnable decoder, which uses a truncated diffusion and a learnable implicit prior.
>
> * The key contribution of DiffFlow (which we refer to as UnifiedDiffFlow in our paper for clarity and to differentiate it from the (Zhang & Chen, 2021)'s DiffFlow) is its unification of multiple generative modeling approaches, providing theoretical guarantees like asymptotic optimality and offering a continuous spectrum of generative models, providing flexibility and theoretical insights into both score-based diffusion models and GANs.
>
>  * DiNof combines deterministic and stochastic mappings to enhance the expressive power of diffusion models, accelerating both forward and backward processes while preserving compatibility with underlying diffusion models.
>     Our model provides extensive experimental evaluations involving a more standard diffusion model without the complexities of integrating a GAN-like discriminator or without the same level of focus on maximal likelihood inference in a unified framework.
>
>
> While we have previously compared our method with TDPM, we further elaborate on this discussion in the paper to better highlight the differences between our approach, TDPM, and Diffflow (Section 2, fifth and sixth paragraphs).
>
> > I strongly suggest authors to add comparisons to [1] in every experiments.
>
> We appreciate the reviewer's suggestion to strengthen the experimental section by including comparisons with TDPM.
> We have compared our method with TDPM in Table 2 on CIFAR-10, and we have incorporated TDPM results on CelebA-HQ-256 (FID=8.38) to Table 3.
> We have also added performance and sampling time results of TDPM (with DDPM++ backbone) on CIFAR-10 (IS=9.62, FID=2.83, Time=37s) to Table 4. These updates will bolster the experimental validation of our approach.

---

> > ### Comment · Reviewer_gvFE · 2024-08-10
> > **The revision addressed my concerns**
> >
> > Thanks for detailed response from authors.  The revised paper with enhanced experiments on TDPM has addressed my critical concerns. I think the current version meets the acceptance criteria of TMLR ( ref: https://jmlr.org/tmlr/acceptance-criteria.html ) Therefore, I vote for acceptance.

---

### Decision · Action_Editor_adDG · 2024-09-18

**Recommendation:** Accept as is

**Comment:**

**General**:
* The paper proposes a new generative model, namely, a combination of a diffusion model and a normalizing flow.  The main idea is to use a data-dependent, deterministic normalizing flow prior as an alternative to the random noise typically used in the context of vanilla diffusion models.

**Pros**:
+ The idea of combining diffusion models with normalizing flows is new and the motivation in theory sounds plausible.
+ There are broad evaluations of experiments to justify the effectiveness of the idea.

**Cons**:
- The idea of combining diffusion models with other existing methods such as GANs, VAEs, or Normalizing flows to speed up training and sampling is not necessarily new. Therefore, a better discussion is needed.
- The experiments are not persuasive enough and miss some relevant baselines. Therefore, new experiments and comparisons are needed.

**Rebuttal**:
* The authors were active in the rebuttal phase.
* They authors updated the paper according to all requests provided by the reviewers. Therefore, the paper could be accepted without further revision rounds.

**Audience:**

The paper's audience greatly overlaps with that of TMLR.

**Claims And Evidence:**

**Claim**:
* The proposed approach enables more accurate modeling of complex distributions and with a smaller number of sampling steps.

**Evidence**:
* The claim is validated on two datasets: CIFAR-10 and CelebA-HQ-256. The proposed approach achieves FID scores of 2.01 and 7.11 on the CIFAR-10 and CelebA datasets, respectively. these scores are on par with SOTA methods. Moreover, the proposed approach results in faster sampling as indicated on chosen datasets.